

# Remote sensing tree classification with a multilayer perceptron

G Rex Sumsion, Michael S. Bradshaw, Kimball T. Hill, Lucas D.G. Pinto and Stephen R. Piccolo

Department of Biology, Brigham Young University, Provo, UT, United States of America

## ABSTRACT

To accelerate scientific progress on remote tree classification—as well as biodiversity and ecology sampling—The National Institute of Science and Technology created a community-based competition where scientists were invited to contribute informatics methods for classifying tree species and genus using crown-level images of trees. We classified tree species and genus at the pixel level using hyperspectral and LiDAR observations. We compared three algorithms that have been implemented extensively across a broad range of research applications: support vector machines, random forests, and multilayer perceptron. At the pixel level, the multilayer perceptron algorithm classified species or genus with high accuracy (92.7% and 95.9%, respectively) on the training data and performed better than the other two algorithms (85.8–93.5%). This indicates promise for the use of the multilayer perceptron (MLP) algorithm for tree-species classification based on hyperspectral and LiDAR observations and coincides with a growing body of research in which neural network-based algorithms outperform other types of classification algorithm for machine vision. To aggregate patterns across the images, we used an ensemble approach that averages the pixel-level outputs of the MLP algorithm to classify species at the crown level. The average accuracy of these classifications on the test set was 68.8% for the nine species.

## INTRODUCTION

As the earth's population grows, understanding global ecosystems is becoming more critical to preserving biodiversity and answering ecological questions. Methods for answering such questions have advanced with improvements in remote-sensing technology. Remote-sensing technology has many applications (*White et al., 2016*); for example, its potential has been demonstrated in forestry and ecology (*Fassnacht et al., 2016*), including forest sustainability assessment (*Foody, 2003*), species identification (*Van Aardt & Wynne, 2007*) and resource inventory evaluation (*Dalponte, Bruzzone & Gianelle, 2012*).

Using remote-sensing technology to improve forest-related species identification is the focus of this study. More specifically, we evaluate the potential to perform remote data retrieval using light detection and ranging (LiDAR) and hyperspectral imagery. Relying

Corresponding author
Stephen R. Piccolo,
stephen_piccolo@byu.edu

on additional data for remote sensing is not a novel idea (*Fassnacht et al., 2016*)—studies have already attempted to use LiDAR and hyperspectral imagery to improve performance (*Dalponte, Bruzzone & Gianelle, 2012*; *Alonzo, Bookhagen & Roberts, 2014*; *Ghosh et al., 2014*). Although previous research has analyzed remote-sensing methods for classification of tree species (*Clark, Roberts & Clark, 2005*; *Castro-Esau et al., 2006*; *Carlson et al., 2007*; *Alonzo, Bookhagen & Roberts, 2014*), more recent developments using neural networks in computer vision technology have yet to be applied to the field (*Ciresan, Meier & Schmidhuber, 2012*; *Krizhevsky, Sutskever & Hinton, 2012*; *Simonyan & Zisserman, 2015*; *Rawat & Wang, 2017*). Supervised machine learning algorithms are well suited to species identification of individual tree crowns, but it is difficult for researchers and practitioners to know *a priori* which algorithm(s) are best for this purpose. We assessed the performance of three common machine-learning algorithms in species identification: the multilayer perceptron (MLP), random forest (RF) and support vector machine (SVM) algorithms. We chose to employ the MLP algorithm due to the recent popularity and success of neural networks in computer-vision applications (*Ciresan, Meier & Schmidhuber, 2012*; *Krizhevsky, Sutskever & Hinton, 2012*; *Simonyan & Zisserman, 2015*; *Rawat & Wang, 2017*). We chose to employ the RF and SVM algorithms because they have had reliable performance in many related studies (*Pal, 2005*; *Pal & Mather, 2005*; *Ghosh et al., 2014*; *Baldeck & Asner, 2014*; *Ferreira et al., 2016*). We hoped this analysis would shed light on differences and advantages of these methods and help us evaluate whether these algorithms have potential to aid in species identification.

We performed this analysis in the context of a community-based challenge. To advance the development of quantitative methods for sampling biodiversity, the National Institute of Standards and Technology (NIST) developed a competition series that allowed researchers to evaluate a common, high-quality, remote-sensing dataset (provided by the National Ecological Observatory Network). By aggregating and evaluating the results from each submission, they hoped to drive the development and identification of optimal models for remote-sensing and species classification. Many other disciplines have accelerated scientific progress through community-based competitions (*Marbach et al., 2010*; *Prill et al., 2011*; *Wan & Pal, 2014*; *Seyednasrollah et al., 2017*). These competitions have helped overcome limitations of individual scientific studies, which often fail to consider a variety of approaches, fail to use common datasets for comparison, and/or use inconsistent evaluation metrics (*Marconi et al., 2019*). We focused on one of the three tasks of this competition: classifying the species or genus of tree crowns from provided hyperspectral and LiDAR data. We used machine-learning classification algorithms to classify species or genus at the tree-crown level. In addition, we evaluated the algorithms' abilities to classify at the pixel level, as done in other studies (*Clark, Roberts & Clark, 2005*; *Castro-Esau et al., 2006*; *Carlson et al., 2007*; *Dalponte et al., 2009*).

## METHODS

### Dataset

All data used in this research is from the Ordway-Swisher Biological Station (OSBS) NEON site (https://www.neonscience.org/data/airborne-data). As provided for the competition,

the data were collected from 23 OSBS NEON distributed plots (1,600 m$^2$) and 20 OSBS NEON tower plots (1,600 m$^2$), which include hardwood/woody wetlands and upland pine/sandhill ecosystems. The data from NEON included the following data products: (1) Woody plant vegetation structure (NEON.DP1.10098); (2) spectrometer orthorectified surface directional reflectance - flightline (NEON.DP1.30008); (3) ecosystem structure (NEON.DP3.30015); and (4) high-resolution orthorectified camera imagery (NEON.DP1.30010). By design of the competition, the tasks of identifying Individual Tree Crown segmentations (ITC) from the images and aligning them with the remote sensing and ground data were separated from the task of classification. Since our task was that of classification, the dataset which we received contained only hyperspectral values (426 bands) and canopy height (from LiDAR) for individual pixels, which were already segmented into individual tree crowns and assigned crown IDs. This ITC data was also aligned with ground data so that each crown ID had a label for both tree species and genus. Some of the crowns had been labeled as "other" to enable researchers to test algorithms' ability to identify species or genera that were not included in the training set. The dataset contained 305 tree crowns for training and 126 for testing. In the training set, there were 9 different labels for species and 5 different labels for genus, including the label "other." The distribution was not equal across all species and genera; 71% of the crowns belonged to the Pinus genera, and 91% of those were of the species 'Pinus palustris'. We used all of the provided features, both the canopy height (LiDAR) and hyperspectral data, to train the implemented algorithms. These features include data associated with the crowns labeled as "other." For additional details about the image data that were provided as part of the competition, we refer the reader to *Marconi et al. (2019)*.

## Classification algorithms

Several prior related remote sensing classification studies used random forests (RF) and support vector machines (SVM) classifiers (*Pal, 2005*; *Pal & Mather, 2005*; *Ghosh et al., 2014*; *Baldeck et al., 2015*; *Ferreira et al., 2016*). Other studies have shown that the multi-layer perceptron (MLP) performs well in many computer-vision applications (*Ciresan, Meier & Schmidhuber, 2012*; *Krizhevsky, Sutskever & Hinton, 2012*; *Simonyan & Zisserman, 2015*; *Rawat & Wang, 2017*); however, this algorithm has been used less frequently for tree-species identification. In contrast to some classification algorithms, the performance of MLP tends to increase with increasing numbers of input variables and data instances (*Ciresan, Meier & Schmidhuber, 2012*; *Simonyan & Zisserman, 2015*; *Rawat & Wang, 2017*). For these reasons, we hypothesized that the MLP may be beneficial for tree classification problems that use both LiDAR and hyperspectral imaging. In this study, we implemented the MLP algorithm and compared its performance against that of the SVM and RF algorithms. The SVM algorithm creates a hyperplane (a barrier) between observations from two classes, attempting to separate the classes by a maximal margin. This margin is adjusted throughout training until classification error is minimized (*Boser, Guyon & Vapnik, 1992*). This algorithm is used frequently when there are complicated patterns in data and has been applied in a wide variety of fields, including remote sensing (*Pal & Mather, 2005*). In addition, it has been applied widely to RNA expression analysis—another biological field

that generates large, complex datasets—where it has been shown to outperform most, if not all, algorithms (*Statnikov et al., 2005*; *Statnikov & Aliferis, 2010*; *Feig et al., 2012*; *Attur et al., 2015*).

RF classifiers operate by creating a number of decision trees. Each decision tree is constructed by identifying variables ("features") in the data that are most informative for dividing the data observations (species or genera, in this case) based on those features, thus forming a branch in the tree. Node creation is repeated iteratively based on the remaining observations. In RF, training occurs via bagging with replacement—that is, random samples are selected from a training set, and trees are fit to that sample subset. The algorithm then classifies test samples on each generated decision tree and outputs a classification for each test sample (*Liaw & Wiener, 2002*; *Pal, 2005*; *Cutler et al., 2007*). This approach theoretically prevents overfitting because the forest is based on a distribution of decision trees consisting of diverse sets of features (*Breiman, 2001*). The MLP, a type of neural network, uses several layers of nodes that fall into one of three categories: input, hidden, or output (*Kuncheva, 0000*; *Haindl, Kittler & Roli, 2007*; *Du et al., 2012*; *Woźniak & Graña, 2014*). A node on a given layer receives a weighted average of the outputs of the previous layer; given its specific weight, that node will contribute to a new weighted average which is propagated down the network until a final output is reached. During training, each node's weights in the network are optimized with the goal of minimizing error through a process called backpropagation (*Hansen & Salamon, 1990*; *Hecht-Nielsen, 1992*; *Buscema, 1998*). The assumption is that a network of nodes can gain insight in supervised learning that a single node cannot.

## Algorithm implementations

Each of the classification algorithms provide many hyperparameters (configuration options) that must be selected. Small changes in these hyperparameters can sometimes lead to drastic performance differences. These settings are often optimized by trial and error—not by structured rules. Some studies have searched to find better ways to optimize certain hyperparameters more effectively, such as the SVM (*Chapelle et al., 2002*), but it is difficult to know which values will be most effective. For this study, to simplify implementation—and to equally compare the previously mentioned algorithms—we employed these algorithms with default parameters (except as noted below) using the scikit-learn Python library (*Pedregosa et al., 2011*). For the RF classifier, we used bootstrapping with replacement and no maximum depth for the decision trees, which allowed the algorithm to continue to add nodes and branches until the algorithm experienced diminishing returns in classification accuracy. For the SVM algorithm, the default settings included a penalty parameter C value of 1.0 and the radial basis function kernel. The MLP algorithm was implemented using three inner layers. Each inner layer was structured to have exactly 40 nodes. We capped the maximum number of training epochs at 200 (the default). Weights were adjusted after each epoch where all training data is passed to the MLP classifier. We chose a learning rate of 0.001 to slowly adjust weights. Both of these adjustments are designed to prevent overfitting. For diagnostic purposes during training, we used the cross_val_score function in scikit-learn (a Python package, https://python.org)

**Figure 1 Visual representation of the ensemble method.** We used the multilayer perceptron algorithm to derive predictions of species and genus based on hyperspectral and LIDAR values at the pixel level. We then aggregated these predictions to crown-level predictions using an ensemble approach that averaged the probabilistic, pixel-level predictions. Acer, AC; Pinus, PI; Quercus, QU; Liquidambar, LI; Unknown, OT.

to evaluate classification accuracy via k-fold cross validation (*Pedregosa et al., 2011*). K-fold cross validation is an algorithm-validation method in which the dataset is randomly split into *k* sections. The validation method iterates k times, allowing a classification algorithm to train using the same model settings in each iteration. In each iteration, these models are trained on all but one of the *k* random sections of the data. The remaining section is used as validation data to evaluate model performance. The *k* iterations allow for each section to have its opportunity to be used in validation. In performing these evaluations, we used the default of 3 folds.

## From the pixel level to the crown level

First, we used the hyperspectral and LiDAR data as inputs to the classification algorithms to classify species and genus for each pixel; next we applied a custom ensemble-averaging method to the pixel-level classifications and used these averaged probabilistic classifications to make tree-crown-level classifications (Fig. 1). This ensemble method (*Dietterich, 2000*) averages the probabilistic classifications across all pixels in a given tree crown. This allows for each input pixel to have a modest impact on the final classifications for the respective tree crowns. An assumption of this approach is that although some individual pixels may be classified incorrectly, a considerable proportion of individual

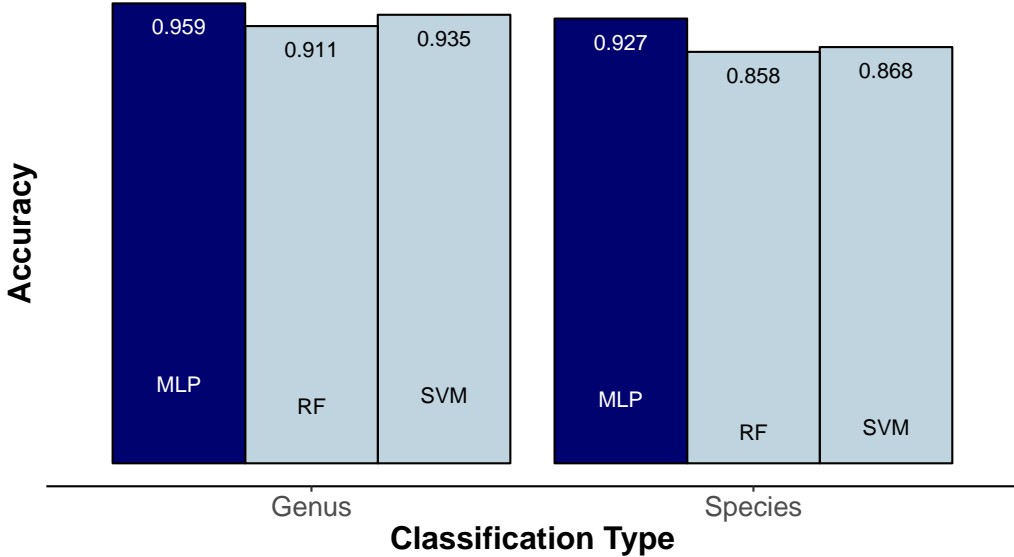

**Figure 2** **Classification accuracy for the classifiers on species (9 labels) and genus (5 labels).** MLP, multilayer perceptron; RF, random forests; SVM, support vector machines.

pixels would be classified correctly, and aggregating evidence across all pixels would lead to correct crown-level classifications. The code that we used in this study is available from https://github.com/byubrg/NIST-Competition-Fall-2017. For details of other methods that were used in the competition, please see the overview paper by *Marconi et al. (2019)*.

**Evaluation metrics**

Through this study, we used rank-1 accuracy to evaluate performance. This method assesses probabilistic classification values and verifies that the highest prediction is the true label. Rank-1 accuracy does not verify the confidence of predictions. Marconi et al. compared all competitors using rank-1 accuracy as a metric. They also used cross-entropy. Cross-entropy evaluates the confidence of predictions. To understand these results in terms of cross-entropy, please refer to *Marconi et al. (2019)*.

## RESULTS

We evaluated three classification algorithms at the pixel level on the training data consisting of hyperspectral and LiDAR data of tree crowns. Our goal was to identify the species or genus of each tree crown. Because many studies in the past only evaluated their algorithms at the pixel level, we aimed to assess the algorithms' ability to make accurate classifications in this context. The SVM and RF algorithms have been used commonly for research-based classification, including for remote sensing, but the MLP algorithm has been used less frequently for this purpose. Overall, the algorithms classified genus and species with high accuracy (85.8–95.9%) and attained higher accuracy for genus than for species (Fig. 2). One reason for these differing levels of accuracy may be that species within a genus are often

## Accuracy of Algorithms Across Genus

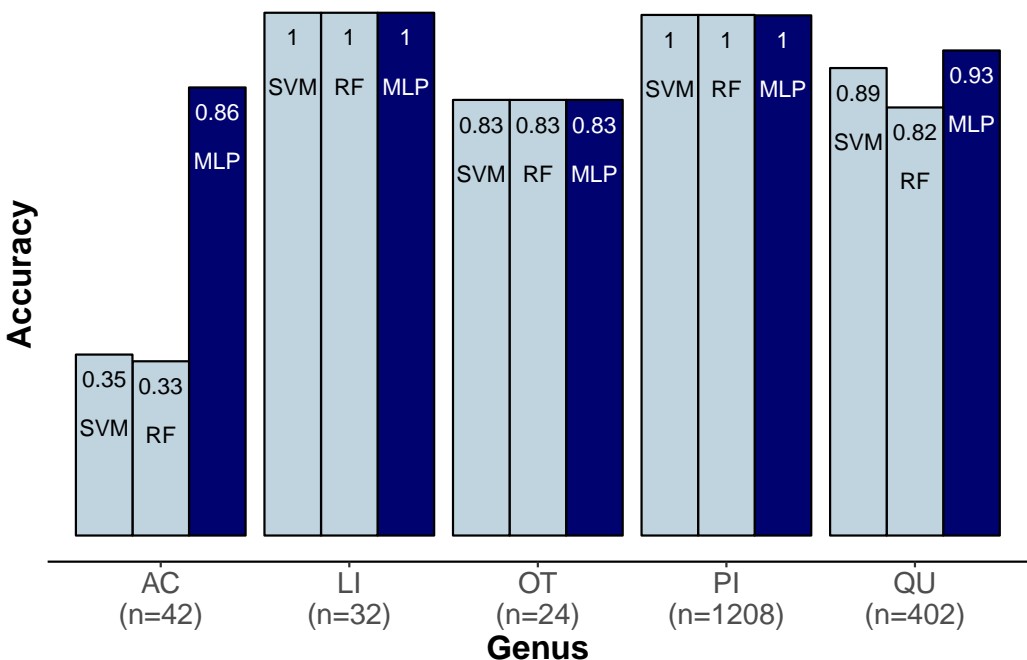

**Figure 3** **Classification accuracy for the classifiers across genera.** Acer, AC; Pinus, PI; Quercus, QU; Liquidambar, LI; Unknown, OT.

quite similar biologically and therefore can also be expected to have similar traits and appear similarly in remote-sensing imagery. For genus-based classification, the algorithms only needed to differentiate among 5 class labels, whereas they needed to distinguish among 9 class labels for species classification. The MLP algorithm's performance dropped least from genus to species (95.9% vs. 92.7%). The SVM and RF algorithms attained classification accuracies of 91.1% and 93.5%, respectively, for genus classifications and 85.8% and 86.8% for species classifications (Fig. 2). MLP also outperformed SVM and RF for all species and genera (Figs. 3 and 4). The differences in performance between MLP and the other algorithms are substantial enough to suggest that the multilayer perceptron should be explored further for tree classification through remote sensing—perhaps especially when using a relatively large number of labels. Our final model used an ensemble-based approach to average pixel-level classifications for the MLP algorithm only. When applied to the competition's test data, our solution obtained an accuracy of 68.8% for crown-level classification (pixel-level classifications were not assessed as part of the final evaluation). Although our solution exceeded the baseline expectation of 66.7% accuracy, our approach failed to generalize well. To better understand these test results and how our results compare to other participants' in the competition, please see the description by *Marconi et al. (2019)*.

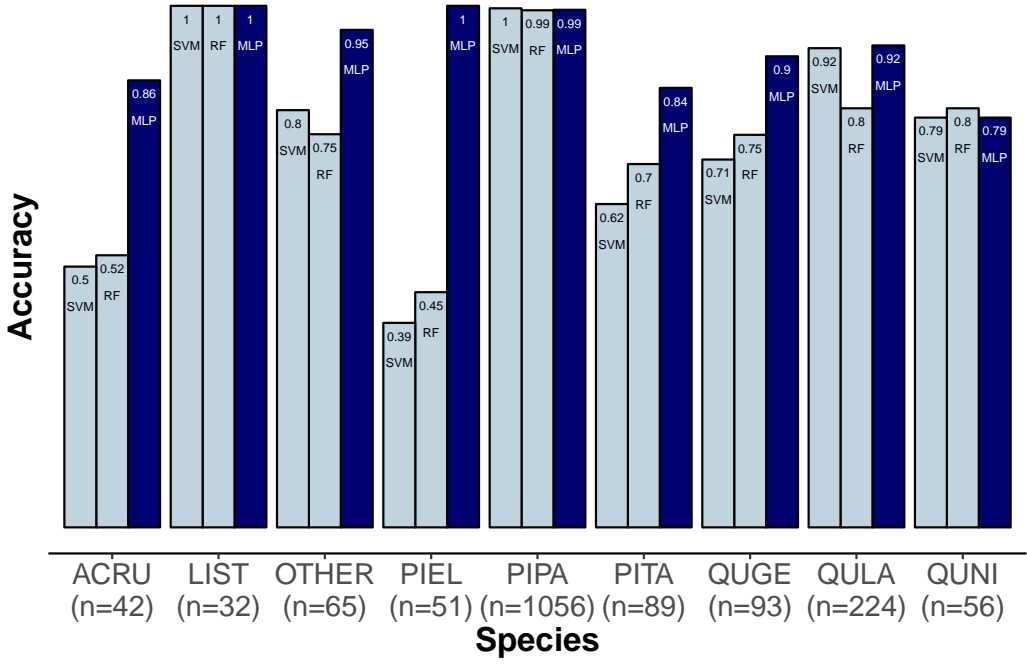

**Figure 4** **Classification accuracy for the classifiers across the species.** Pinus palustris, PIPA; Quercus laevis, QULA; Liquidambar styraciflua, LIST; Pinus elliottii, PIEL; Quercus nigra, QUNI; Acer rubrum, ACRU; Unknown species, Other; Quercus geminata, QUGE; Pinus taeda, PITA.

## DISCUSSION

As early as 1998, computer-vision techniques have been used to answer biological questions. In some of these studies, hyperspectral imagery has been used to differentiate between similarly colored items, such as *chlorophyll a* and *chlorophyll b* (*Blackburn, 1998*). Subsequent studies have used computer-vision techniques to enhance tree species classification. While employing statistical-learning algorithms, one of these first studies evaluated accuracies of pixel-level classifications on the leaf and crown scales (*Clark, Roberts & Clark, 2005*). In subsequent years, additional researchers assessed the effectiveness of using hyperspectral wavelengths in images. These studies found a correlation between higher classification accuracy and the use of more wavelengths (*Castro-Esau et al., 2006*; *Carlson et al., 2007*). To our knowledge, *Dalponte et al. (2009)* were first to evaluate the use of hyperspectral imagery in remote sensing. Since that time, many studies have developed even more effective algorithms and forms of data representation for remote tree-species classification. One of the most interesting of these studies combined hyperspectral and LiDAR data to obtain higher accuracies (*Alonzo, Bookhagen & Roberts, 2014*), similar to the approach used in our study.

Further building on the remote-sensing analyses performed in these previous studies, we evaluated two classification algorithms (SVM and RF) that have been used more

traditionally in remote-sensing applications, as well as the MLP algorithm, a basic neural network. Only in recent years has the success of neural-network models been demonstrated in a wide variety of computer-vision problems due to increased parallel processing speed in GPUs (*Ciresan, Meier & Schmidhuber, 2012*). Reducing the need for feature engineering, neural networks can integrate different types of data, such as hyperspectral images, RGB images, and LiDAR readings. Our results are congruent with many others who have used neural-network models to achieve higher performance on computer-vision tasks. Our study provides support for additional research on the use of neural-network algorithms in remote image sensing applications.

## CONCLUSION

Of the models that were tested in our comparison, the MLP algorithm consistently demonstrated superior performance. The results of our comparison add to the body of research demonstrating the relatively high accuracy of neural-network based algorithms in computer-vision classification problems (*Simonyan & Zisserman, 2015*; *Rawat & Wang, 2017*). Neural networks, including the MLP algorithm, frequently outperform other classification methods and present additional advantages including a fast learning rate and the ability to stack layers in the network to represent complex relationships (*Ciresan, Meier & Schmidhuber, 2012*). Specifically, our results suggest the MLP algorithm is an effective method for tree classification using hyperspectral and LiDAR imagery.

The performance gap between MLP and the other algorithms is most clearly shown in the models' evaluation when classifying individual pixels, using the label of their corresponding crown. However, when aggregating that evidence across pixels-level classifications, the accuracy of our models decreased considerably. This drop could be due to oversimplification of our ensemble method in that it did not account for spatial relationships among the pixels and did not correct for outlier effects. Alternative approaches that may have led to better results include (1) using a convolutional neural network to aggregate the pixel-level classifications and account for spatial relationships (*Krizhevsky, Sutskever & Hinton, 2012*), (2) using an ensemble method that is more robust to outliers (*Kuncheva, 0000*; *Haindl, Kittler & Roli, 2007*; *Du et al., 2012*; *Woźniak & Graña, 2014*), (3) including all three classification algorithms in our ensemble, thus potentially reducing the effect of outliers and incorrect pixel-level classifications, and/or (4) attempting to optimize our model's hyperparameters.

## ACKNOWLEDGEMENTS

This material is based in part upon work supported by the National Science Foundation through the NEON Program. We also acknowledge the Brigham Young University Fulton Supercomputing Laboratory for computational resources that were used for this analysis.

### Funding

The ECODSE competition was supported, in part, by a research grant from NIST IAD Data Science Research Program to DZ Wang, EP White, and S Bohlman, by the Gordon and Betty Moore Foundation's Data-Driven Discovery Initiative through grant GBMF4563 to EP White, and by an NSF Dimension of Biodiversity program grant (DEB-1442280) to S Bohlman. The National Ecological Observatory Network is a program sponsored by the National Science Foundation and operated under cooperative agreement by Battelle Memorial Institute. The funders had no role in study design, data collection and analysis, decision to publish, or preparation of the manuscript.

### Grant Disclosures

The following grant information was disclosed by the authors:
NIST IAD Data Science Research Program.
Gordon and Betty Moore Foundation's Data-Driven Discovery Initiative: GBMF4563.
NSF Dimension of Biodiversity: DEB-1442280.
National Science Foundation.
Battelle Memorial Institute.

### Competing Interests

The authors declare there are no competing interests.

### Author Contributions

- G Rex Sumsion conceived and designed the experiments, performed the experiments, analyzed the data, prepared figures and/or tables, authored or reviewed drafts of the paper, approved the final draft.
- Michael S. Bradshaw conceived and designed the experiments, performed the experiments, authored or reviewed drafts of the paper, approved the final draft.
- Kimball T. Hill and Lucas D.G. Pinto conceived and designed the experiments, authored or reviewed drafts of the paper, approved the final draft.
- Stephen R. Piccolo analyzed the data, authored or reviewed drafts of the paper, approved the final draft.

### Data Availability

  GitHub: https://github.com/byubrg/NIST-Competition-Fall-2017.

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
