# Peer review of "Remote sensing tree classification with a multilayer perceptron"

_PeerJ, doi:10.7717/peerj.6101_

## Round 0.1 · original submission · Minor Revisions

· Academic Editor

Minor Revisions

I enjoyed reading this paper comparing several methods for tree classification. As both of the reviewers noted, though, the paper should be expanded to add context, especially the intro and discussion. There were also numerous grammatical/terminology issues and missing citations.

Reviewer 1 ·

Basic reporting

The manuscript is generally clear but is a bit too concise. It would be good to include in the Introduction a bit of background regarding current progress and challenges in the field of spectral classification of trees. The study and results are represented well by the figures, but I recommend minor edits. For Figure 1, include in the figure caption the genera that are abbreviated. For Figure 2, include two more panels to illustrate accuracy within genus and species (variation by genus and species is of interest, as some genera and species might be more accurately classified than others).

Experimental design

A few important details are missing from Methods section. There is no information regarding the tree dataset (how many samples per species) and any data splitting (for training/testing of classifications). The reader also needs the information regarding evaluation metrics used in the competition (rank accuracy and cross-entropy).

Validity of the findings

The accuracy values are a bit difficult to assess in terms of significance because they are provided at the genus level and species level as averages, I think. I expected genus (5) and species (9) specific accuracy values to be reported, in two matrices (for genus level and for species level analysis). The Discussion section is too short and does not go into the details of the results of this study. It needs to be further developed to highlight the findings of this study and their importance in the context of previous work in the field of tree identification from hyperspectral imagery.

Additional comments

The manuscript can be strengthened by expanding the Introduction and Discussion sections.
I noticed that the authors use “predicted” instead of “classified” when referring to identifying tree genera or species. I suggest revising the use of predicted, predictions, etc.
Abstract:
Line 15: insert “two” before algorithms
Line 16: spell out MLP acronym
Line 21: is 68.8% the average accuracy value, over all 9 species? Please clarify.
Introduction:
Lines 23-27: citations needed in this paragraph
Line 40: “task III” is not relevant in this context, could revise as “one of the tasks”; also, need to include here a paragraph about current progress and problems in the field of tree mapping from hyperspectral data
Line 41: lidar stands for “light detection and ranging”; also here and elsewhere in the manuscript, use LiDAR or lidar spelling (not LIDAR).
Materials and Methods:
Line 50: mention location of NEON data (repository, if readers want to download these data).
Line 57: Marconi et al citation is missing publication year.
Line 58: the section Algorithm Training can me merged into the next section, Classification Algorithms.
Line 59: mention that the function is python
Line 65: cite “Other studies”
Line 71: Scholkopf et al is missing publication year
Line 72: the word frequently is repeated in this sentence
Lines 75-80: citations needed in this paragraph
Line 76: “k features” is unclear to me – what are features in this context?
Line 80: “each tree in the forest” – the use of forest in this context is ambiguous; I think it refers to plants (trees), not classification algorithm; then could be reworded as “each tree species in the forest”
Lines 85-89: this paragraph needs citations
Lines 99-104: this explanation is a bit abstract – is it possible to explain (briefly) the implications of these parameters?
Line 117: Marconi et al citation is missing publication year; same thing in line 141 (Results(
Discussion
Needs to be further developed; discuss the significance of your results; include the context of previous research. Also, remove “wasn’t” (lines 146 and 151), and reword “hyperspectral images provide the highest accuracy” because it seems vague (highest among other types of imagery?)
line 270: reword “showed different within…”
Conclusions: line 334: reword “wasn’t”

Reviewer 2 ·

Basic reporting

In this manuscript the authors compared two traditional classification algorithms (SVM,RF) with a neural-network based method (MLP). The authors have also implemented a clever way of aggregating the pixel-level classification to the tree crown level using an ensemble-based method. This should be of value for other remote sensing applications as well.

The introduction section is very brief and do not provide enough context to the study. For example, the authors can discuss previous literatures on species classification, on hyperspectral and LiDAR, and highlight what has been done, what are the key issues and how the current work tries to solve it. The authors can also discuss here why the specific algorithms were selected for this work (SVM, RF, and MLP).

Experimental design

The authors compared the RF and SVM classification algorithms with the MLP – yet have nowhere in the manuscript described what the MLP algorithm is or cite any related literature that does so.

Validity of the findings

The manuscript reports cross-validation classification accuracies for species and genera (Figure 2) but does not report how the algorithms performed with the test samples (30% of the samples were provided as the test by the competition).

I feel that the discussion section can be improved.

Additional comments

Lines 58-61: Please briefly describe how k-fold cross validation works.
Line 65. Please cite which “other studies” that have shown MLP is better for tree classification.
Line 71: Missing year in Scholkopf et al.
Lines 69-74: SVM is also widely used in patterns in remote sensing data! It would be more relevant and interesting to readers to cite the literatures that applies SVM in remote sensing/tree classification than the application in RNA expression
Line 91. Define ‘hyperparameters’
Line 117. Missing year in Marconi et al.

---

## Round 0.2 · accepted · Accept

· Academic Editor

Accept

Thanks you for your thoughtful attention to the reviewers' comments. I believe your work is an excellent contribution to the collection of papers in response to the NIST-NEON remote-sensing classification challenge.

#